# Health-Related Quality of Life Norm Data of the Peruvian Adolescents: Results Using the EQ-5D-Y

**DOI:** 10.3390/ijerph18168735

**Published:** 2021-08-19

**Authors:** Roxana Paola Palacios-Cartagena, Jose Carmelo Adsuar, Miguel Ángel Hernández-Mocholí, Jorge Carlos-Vivas, Sabina Barrios-Fernández, Miguel Angel Garcia-Gordillo, María Mendoza-Muñoz

**Affiliations:** 1Promoting a Healthy Society Research Group (PHeSO), Faculty of Sport Sciences, University of Extremadura, 10003 Cáceres, Spain; ropalacio@alumnos.unex.es (R.P.P.-C.); jorgecv@unex.es (J.C.-V.); mamendozam@unex.es (M.M.-M.); 2Physical Activity and Quality of Life Research Group (AFYCAV), Faculty of Sport Science, University of Extremadura, 10003 Cáceres, Spain; mhmocholi@unex.es; 3Social Impact and Innovation in Health (InHEALTH), University of Extremadura, 10003 Cáceres, Spain; sabinabarrios@unex.es; 4Universidad Autónoma de Chile, Sede Talca 3467987, Chile; miguel.garcia@uautonoma.cl

**Keywords:** adolescents, EQ-5D-Y, health conditions, health-related quality of life (HRQOL), health status assessment

## Abstract

**Simple Summary:**

Health-related quality of life in the adolescent stage is of vital importance because it provides the adolescent with an impression of his or her functional capacity, allowing him or her to examine, discover, and distinguish sensations and concerns. This indicator provides information on the person’s state of health. The EQ-5D-Y questionnaire was used to measure health-related quality of life, due to its simplicity and ease of use. Therefore, the aim of the present study was to describe the perceived quality of life in Peruvian school adolescents. Despite the increase in health-related quality of life studies, there is little research on the Latin American population. For this reason, it would be important to have normative data in Peru in order to estimate the impact on health-related quality of life in adolescents, since it allows comparisons of health-related quality of life between the general population and the pathological population. The results have shown that there are differences in the EQ-5D-Y between ages, as well as between weight groups established according to body mass index. In conclusion, adolescents have perceived favorable health-related quality of life sensations and the EQ-5D-Y has shown to be a feasible and useful questionnaire, in addition to having internationally recognized and validated cross-cultural characteristics.

**Abstract:**

(1) Introduction: There is a growing interest in health-related quality of life (HRQOL) in adolescent population. The EQ-5D-Y is a generic HRQOL instrument that allows adolescents to understand the health status of different levels of physical, mental, and social health. This study was carried out with an adolescent population in Peru. The main objective of this article is to report the normative values of the EQ-5D-Y questionnaire in Peruvian adolescents. (2) Methods: The EQ-5D-Y questionnaire was administered to Peruvian adolescent students. A total of 1229 young people participated in the survey. The EQ-5D-Y score was reflected as a function of sex and age. (3) Results: The mean utility index of the EQ-5D-Y for the total sample was 0.890; this rating was significantly better for males at (0.899) and females at (0.881). The ceiling effect was higher for adolescent males with (47.3) females (40.7). (4) Conclusions: The results of the present study provide evidence that schooled adolescents show a positive perception of HRQOL.

## 1. Introduction

In recent decades, the public health sector has seen a growing interest in research and evaluation of health-related quality of life (HRQOL), as an indicator of physical and mental well-being in children and adolescents, providing information on morbidity and mortality measures in public health focused on symptoms and signs of disease rather than subjective assessment of the child/adolescent’s functioning and well-being [1].

When a HRQOL’s questionnaire is applied in adolescents, it allows them to have a perception about their functional capacity, in which they can observe, detect, and discriminate sensations and concerns. This indicator allows them to discover different aspects of health, at different levels of physical, psychological, and social well-being of young people [2], due to the fact that the quality of life at this stage is closely related to the quality of life in adulthood. For these reasons, the components of HRQOL should receive special attention in pediatric and adolescent care [3].

The measurement of HRQOL requires questionnaires that can be administered quickly and simply, and allow reliable and valid results to be received. Several questionnaires with these characteristics exist today. From its conception, the EuroQol-5D (EQ-5D-3L) was created as a generic and standardized questionnaire, which is simple to answer. It also includes a visual analogue scale (VAS) where the respondent rates his or her own general health on a scale ranging from 0 (worst imaginable health) to 100 (best imaginable health); this is known as the EQ-VAS value and directly reflects the HRQOL as rated by the respondents. This instrument is widely used throughout the world and is available in more than 170 languages [4]. It is feasible to assess the HROoL of the population, given that it is easy to use and intended to provide valid and reliable information [5]. The EuroQol Group perceived a growing interest in assessing HRQOL in younger people, which was a suitable measure for children and adolescents, creating the EQ-5D-Y instrument. This was developed on behalf of the EuroQol Group in the year 2009/2010 by a team of researchers from 7 countries (Germany, Italy, South Africa, Spain, Sweden, Holland, and United Kingdom) [6]. This version for children and young people is the most internationally used HRQOL’s questionnaire among children and adolescents [7].

Data obtained from this questionnaire in a general population (normative values) allows comparisons of HRQOL with other pathological populations, providing information on specific domains that differ between subjects with or without symptoms to address during treatment or compare patient profiles with particular conditions, including similar age group or gender, aiding health policy development and planning [8]. 

There are numerous publications on the normative data of the EQ-5D-5L and EQ-5D-3L questionnaire among adults in European countries such as Spain [9], Denmark [10], Germany [11], Portugal [12], and United Kingdom [13]. In another continents, there are publications in Japan [14], Australia [15], Hong-Kong [16], and Russia [17]. However, the establishment of normative values in children/adolescents has only been published in Japan [18]. These data can be used in cost-effectiveness analyses, as they are based on healthy individuals. In addition, it should be noted that the population norms for children/adolescents (obtained using the EQ-5D-Y) and adults (obtained using the EQ-5D or EQ-5D-5L) are likely to differ, so it is important to have reference data for each population.

Despite the increase in HRQOL studies, there are few studies aimed at the Latin American population, particularly in an epidemiological and instructional setting [19]. Therefore, having normative data of children/adolescents in Peru could be of interest to estimate the impact on HRQOL in adolescent school students as a useful resource to interpret self-reported outcomes, compare HRQOL assessment results within different populations, determine health deviations, measure inequalities, as well as improve health care, and guide clinicians in planning specific interventions. To our knowledge, there is no study in which the EQ-5D-Y instrument has been used in children/adolescents in Peru, therefore, the aim of this study was to describe the perceived quality of life in Peruvian schooled children/adolescents.

## 2. Materials and Methods

### 2.1. Design

A single-measure cross-sectional study was conducted.

### 2.2. Ethical Approval

Ethical approval was granted by the bioethics and biosafety committee of the University of Extremadura on 10 December 2020 (approval number: 162/2020), in accordance with the updates of the Declaration of Helsinki, as modified by the 64th General Assembly of the World Medical Association. Association (Fortaleza, Brasil, 2013) and the Law 14/2007 on Biomedical Research.

### 2.3. Partipants

Data collection was performed in school or extracurricular sports activities. Through the cell phone, students accessed the survey link by completing the questionnaire. All participants met the following inclusion criteria: (1) age between 12 and 18 years; (2) informed consent signed by parents or legal guardians; and (3) acceptance of the participant in the study.

The total sample consisted of 1229 adolescent students, of which 622 were female (50%) and 607 were male (50%), aged 12 to 18 years with a mean age of 14.62 (±1.64). The follow figure (Figure 1) shows the recruitment process. 

### 2.4. Measures

The sociodemographic characteristics collected in the survey were age, sex, weight, height, and educational level. 

### 2.5. Instrument

The EQ-5D-Y is a self-administered questionnaire intended to measure the self-perceived health status of adolescents. The Spanish version of the EQ-5D-Y [7] questionnaire and its Proxy version [20] comprise a descriptive section of five dimensions (mobility, self-care, usual activities, pain/discomfort, and anxiety/depression) and each one contains three levels where the person must indicate the severity according to his or her health status at the time of completing the questionnaire [21]. This measure has been tested in various pathologies, and has been able to prove the validity of the instrument in certain areas and track the progress of patients (children and adolescents) through an illness or treatment [22]. Therefore, it can be considered a short instrument, simple to complete, and generic, as it does not refer to any specific disease [23]. 

### 2.6. Statistical Analysis

All information collected was tabulated in a database designed specifically for this study. Statistical analyses were carried out using IBM SPSS Statistics (Version 25, IBM SPSS, Chicago, IL, USA) and personal data were kept anonymous.

Data were presented as the median and interquartile range both for the total sample and segmented by sex, age, and bodyweight category. Normality and homogeneity were tested using the Kolmogorov–Smirnov test and Levene’s test, respectively.

The Mann–Whitney U test was applied to analyze between sex differences. Chi-square was used for multiple-comparisons in categorical variables (age and bodyweight category).

The ceiling effect was calculated as the percentage of participants who obtained the best possible health status (11111).

The EQ-5D-Y utility index was computed using a regression model according to valuation study specifications reported by Badia, et al. [24].

## 3. Results

Table 1 shows the main characteristics of the study. A total of 1229 Peruvian adolescents participated in the study. Of these, 622 were female and 607 were male. The EQ-5D-Y utility index for the total sample was 0.890. This score was significantly significant (*p* = 0.027) for males (0.899) than for females (0.881). The ceiling effect was higher for men (47.3%) than for women (40.7%). However, the ceiling effect experienced a reduction as the age of adolescent students increased with the exception of 17 years old. In addition, statistically significant differences were detected in the EQ-5D-Y utility index between ages, as well as between weight groups established according to BMI.

We can observe in Table 2 that the EQ-5D-Y utility index of the sample segmented by sex, according to age and BMI category. When analysing the results by age groups, it can be seen that 13-year-old adolescents reported the highest mean utility at 0.931. However, the lowest group included 12-year-old males with a mean of 0.886. The 12-year-old females showed a mean of 0.908, being the highest; and the 15 year old group showed a mean of 0.842, being the lowest in the female age group. It can also be observed that the highest ceiling effect is found at the age of 12 years in both sexes: female 69 among females (68.3) and 60 among males (60.6). 

As for BMI (Table 2), we can observe, according to the classification of their assigned category, that males are in their normal weight compared to females (356 vs. 228); however, in overweight it shows us a number of females (309) higher than males (218). As for obesity, we observe 83 females versus 29 males who present obesity. There is a total of two underweight males and two underweight females. It can also be observed that there is a statistically significant difference in sex for each weight category in the EQ-5D-Y utility index.

In Table 3, we can observe similar gender difference for each dimension. In the mobility dimension, 95.7% of the males and 94.9% of the females responded “I have no problem walking”. In the self-care dimension, 95.1% of men and 92% of women responded “I have no problems washing or dressing myself”. In the usual activities dimension, 78.1% of the men and 73.8% of the women responded “I have no problem performing my usual activities”. Regarding the pain/discomfort dimension, 69.2% of men and 67% of women responded “I have no pain or discomfort”. Finally, in the anxiety/depression dimension, 52.70% of the men and 44.1% of the women responded “I am neither anxious nor depressed”. We observed a high percentage in level two of the anxiety and depression dimension in women with a percentage of 53.3% and men with 47.4%. In the dimension of pain/discomfort, similar results are not shown for both sexes: 30.1% for men and 30.4% for women. In habitual activities, we observed a percentage of 43% for men and 51% for women. Regarding the dimensions of self-care and mobility, both sexes registered a percentage below 5%.

Table 4 shows the distribution of Peruvian adolescents according to their state of health. A total of 540 young people (representing 43.9% of the total sample) reported a perfect state of health 11111. The second most common health status was 11112, with a total of 233 adolescents, 18.9% of the sample having no problems with mobility, self-care, usual activities and anxiety/depression, but slight pain or discomfort.

## 4. Discussion

This study involved 1229 adolescents of both sexes at a secondary school age from the city of Lima. To the best of our knowledge, this is the first article that aims to provide normative data on the EQ-5D-Y in Peruvian adolescent school students.

The main finding of this study was to analyze the normative values of the EQ-5D-Y questionnaire in Peruvian schooled adolescents. Our results indicate that this self-report instrument was feasible for adolescent students aged 12 to 17 years. The EQ-5D-Y was employed in healthy schoolchildren in the general population of Sweden [25], and it was also applied in students in the city of Cape Town in South Africa to assess HRQOL in adolescents [26]. Using population health surveys in adolescents allows us to access norms of health perception references [6]. 

Regarding HRQOL, in relation to sex, adolescent males in our study show better quality of life. These results are similar to those obtained by Quiceno and Vinaccia [27], who evaluated Colombian children and adolescents and reported that males perceive better standards of living than females. Similarly, with respect to gender differentiation, males generally showed a better self-assessment of their physical well-being compared to females [28].

We also found a low incidence of serious problems reported in the different dimensions; results that are in agreement with the child/youth population of European [20,29], South Africa and Australia countries [7].

In terms of age, higher frequency is observed in the dimensions of “pain/discomfort” and “feeling sad/worried” consistently in both sexes as age increases. These results are consistent with the validation study of the EQ-5D-Y in a Spanish school [20].

A large percentage of participants reported that they had no problems in all dimensions of our investigation with an EQ-5D-Y health profile of 11111, comprising almost 50% of our sample. This finding is consistent with previous studies used with the EQ-5D-Y. For example, 81.2% of Korean respondents reported the highest health status (11111) [30]. Likewise, in the study on the EQ-5D-Y health states in the general population in Sweden, 63.4% of respondents reported a full health status [25]. In another study on children and adolescents with asthma, the results showed that 48.9% of the respondents reported this same health status [25]. A study by [31] indicates that this may be due to the ceiling effect, because EQ-5D-Y may not be able to distinguish when the participant’s health is close to full health.

In this study, we observed that overweightness and obesity are prevalent and negatively affect HRQOL. We observed a high percentage of overweight adolescents, i.e., 218 males and 309 females, and obesity in 29 males and 83 females. Our data agree with the Demographic and Family Health Survey (ENDES) of the National Institute of Statistics and Informatics (INEI), which indicated that 58% of Peruvians over 15 years of age suffer from overweight and obesity. Similarly, in the study by Lozano-Rojas [32], where the same ages were taken as in our study, they indicated a high prevalence of overweightness. Therefore, approximately one out of every two or three schoolchildren in the studied population present obesity or overweightness. The results found are almost similar to the prevalence of obesity and overweightness in adolescents aged 12 to 19 years in Ecuador [33].

On the other hand, normative data in specific populations, such as adolescents, are fundamental because they allow comparisons of HRQOL between pathological populations and the general population, which helps in the development and planning of health policies [34,35]. Furthermore, in the area of research, since normative data allow to assess the clinical relevance of specific treatments and interventions, they can be a useful resource for the interpretation of the outcome already indicated by the subject [36].

The finding that men have a better quality of life is very significant. Surely there are multiple reasons that explain why females have a worse quality of life than males. In this sense, a hypothesis could be proposed according to a publication prepared by the Peruvian Ministry of Education. The document points out that violence in Peru is a major social problem in adolescence (most victims are between 14 and 18 years old), with a higher prevalence in girls than in boys [37].

According to an article by Romero et al. [38], all violence, especially sexual violence, has a psychological impact. This could be a possible explanation for why girls/adolescents have worse utility values in the anxiety/depression dimension of the EQ-5D-Y.

Finally, we can confirm that, although further studies are needed to establish the cross-cultural equivalence of the instrument, the results obtained represent an important starting point for measuring HRQOL in the Peruvian population.

There is no information on the performance of EQ-5D-Y in specific populations; it was not possible to obtain additional clinical data. 

## 5. Conclusions

In conclusion, it was observed that Peruvian schooled adolescents have a favourable perception of HRQOL. Males showed better health than females. It is also observed that, as age increases, there seems to be worse HRQOL, which may be due to the physical and psychological changes that occur at this stage. As for the psychometric properties, the questionnaire presents certain problems of high ceiling effect. In order for girls to improve their health-related quality of life data, it may be of interest to include educational programmes that raise awareness of the specific issues for girls that lead to higher levels of anxiety and depression. These programmes should not only target students but also parents, family members, and the wider educational community. It would be in the interest of the political authorities to adopt concrete policies and laws to enable girls to improve their health-related quality of life.

## Figures and Tables

**Figure 1 ijerph-18-08735-f001:**
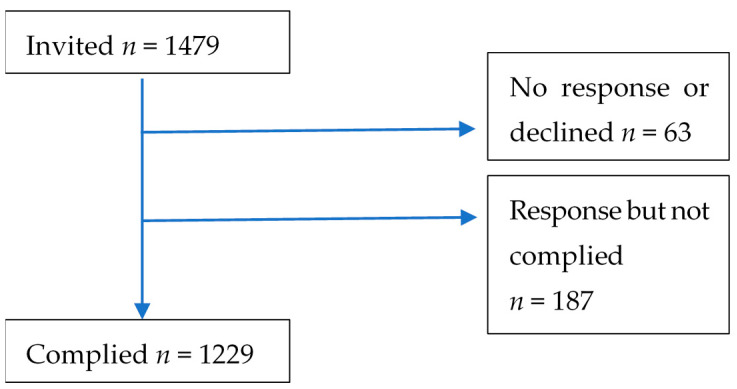
Recruitment process.

**Table 1 ijerph-18-08735-t001:** Sample characteristics. EQ-5D-Y adolescents’ population normative values.

			EQ-5D-Y Utility Index					Ceiling Effect	
	*n*	(%)	Mean	SD	Median	RI	*p*	*n*	(%)
Total	1229	100	0.890	0.165	0.938	0.152		540	43.9
Gender									
Female	622	50.6	0.881	0.176	0.938	0.151	0.027 ^a^	253	40.7
Male	607	49.4	0.899	0.154	0.938	0.151		285	47.3
Age									
12	200	16.3	0.897	0.201	1	0.890		129	64.5
13	200	16.3	0.917	0.150	1	0.151		117	58.5
14	200	16.3	0.886	0.175	0.938	0.151	<0.001 ^b^	88	44.0
15	200	16.3	0.857	0.187	0.938	0.222		59	29.5
16	212	17.2	0.896	0.107	0.938	0.151		62	29.2
17	217	17.7	0.885	0.157	0.938	0.222		85	39.2
IMC Category									
Low weight	6	5	0.905	0.102	0.938	0.222		2	33.3
Normal weigh	584	47.5	0.892	0.149	0.938	0.151		225	38.5
Overweigh	527	42.9	0.904	0.155	1	0.151	<0.001 ^b^	264	50.1
Obesity	112	9.1	0.811	0.254	0.938	0.285		49	43.8

^a^*p*-value from the Mann-Whitney U test; ^b^ Chi-square test.

**Table 2 ijerph-18-08735-t002:** Study sample characteristics. EQ-5D-5L adolescents’ population normative values by gender.

			EQ-5D-Y Utility Index									Ceiling Effect	
	Male	Female	Male				Female					Male	Female
	*n* (%)	*n* (%)	Mean	SD	Median	RI	Mean	SD	Median	RI	*p* *	*n* (%)	*n* (%)
Age													
12	99 (16.3)	101 (16.2)	0.886	0.207	1	0.151	0.908	0.195	1	0.665	0.262	60 (60.6)	69 (68.3)
13	100 (16.5)	100 (16.1)	0.931	0.125	1	0.972	0.904	0.171	1	0.151	0.355	61 (61)	56 (56)
14	100 (16.5)	100 (16.1)	0.888	0.176	0.938	0.151	0.884	0.175	0.938	0.151	0.896	41 (41)	47 (47)
15	100 (16.5)	100 (16.1)	0.871	0.153	0.938	0.151	0.842	0.215	0.938	0.222	0.548	31 (31)	28 (28)
16	100 (16.5)	112 (18.0)	0.900	0.119	0.938	0.151	0.893	0.944	0.938	0.151	0.134	40 (40)	22 (19.6)
17	108 (17.8)	109 (17.5)	0.916	0.119	0.969	0151	0.853	0.182	0.938	0.222	0.002	54 (50)	31 (28.4)
IMC category													
Low weight	4 (7)	2 (3)	0.888	0.128	0.889	0.222	0.938	0	0.938	0	1	2 (50)	0 (0)
Normal weigh	356 (58.6)	228 (36.7)	0.899	0.135	0.938	0.151	0.880	0.168	0.938	0.151	0.151	147 (42.3)	78 (34.2)
Overweigh	218(35.9)	309 (49.7)	0.908	0.168	1	0.133	0.902	0.146	0.938	0.151	0.014	127 (58.3)	137 (44.3)
Obesity	29 (4.8)	83 (13.3)	0.837	0.232	0.938	0.222	0.802	0262	0.938	0.462	0.975	11(37.9)	36 (45.8)

* *p* for Mann-Whitney U test.

**Table 3 ijerph-18-08735-t003:** Percentage frequency distributions of EQ-5D-Y dimensions by gender and age group.

	Mobility			Self-Care			Usual Activities			Pain/Discomfort			Anxiety/Depression		
Level	Total	Male	Female	Total	Male	Female	Total	Male	Female	Total	Male	Female	Total	Male	Female
All															
1	95.2	95.7	94.9	93..4	95.1	92	75.9	78.1	73.8	68	69.2	67	48.3	52.7	44.1
2	4.7	4.3	5.1	6.3	4.9	7.7	23.8	21.7	25.9	31.6	30.6	32.6	47	43	51
3	0	0	0	2	0	3	2	2	3	2	2	3	4.6	4.3	5
12															
1	92	90.9	93.1	92.5	92.9	92.1	79.5	79.8	79.2	75	70.7	79.2	69.5	67.7	71.3
2	8	9.1	6.9	7.5	7.1	7.9	20.5	20.2	20.8	25	29.3	20.8	23	23.2	22.8
3	0	0	0	0	0	0	0	0	0	0	0	0	7.5	9.1	5.9
13															
1	96	97	95	96	99	93	82	84	80	74.5	77	72	62.5	66	59
2	4	3	5	3.5	1	6	18	16	20	25.5	23	28	34	31	37
3	0	0	0	5	0	1	0	0	0	0	0	0	3.5	3	4
14															
1	96.5	95	98	94	95	93	78	78	78	69.5	72	67	45	43	47
2	3.5	5	2	6	5	7	22	22	22	30	27	33	49	52	46
3	0	0	0	0	0	0	0	0	0	5	1	0	6	5	7
15															
1	94	97	91	88.5	93	84	72	76	68	61.5	59	64	34.5	36	33
2	6	3	9	11	7	15	27	24	30	37.5	41	34	59.5	58	61
3	0	0	0	5	0	1	1	0	2	1	0	2	6	6	6
16															
1	96.2	95	97.3	96.2	94	98.2	73.1	73	73.2	63.7	64	63.4	36.8	50	25
2	3.2	5	2.7	3.8	6	1.8	26.9	27	26.8	36.3	36	36.6	62.3	49	74.1
3	0	0	0	0	0	0	0	0	0	0	0	0	9	1	9
17															
1	96.8	99.1	94.5	93.5	96.3	90.8	71.4	77.8	65.1	65	72.2	57.8	42.9	53.7	32.1
2	3.2	9	5.5	6.5	3.7	9.2	28.1	21.3	34.9	35	27.8	42.2	53	44.4	61.5
3	0	0	0	0	0	0	5	9	0	0	0	0	4.1	1.9	6.4

**Table 4 ijerph-18-08735-t004:** Distribution of health status.

EQ-5D-Y Health Status	Frequency (*n*)	Valid Percentage (%)	Accumulative Percentage (%)
11111	540	43.9	43.9
11112	233	18.9	62.8
11222	151	12.3	75.1
11122	100	8.1	83.3
11212	36	2.9	86.2
11223	29	2.4	88.5
22222	26	2.1	90.7
11121	25	2.0	92.7
12222	12	1.0	93.7
22223	11	0.9	94.6
11211	9	0.7	95.3
12112	6	0.5	95.8
12122	6	0.5	96.3
12223	5	0.4	96.7
11221	4	0.3	97.0
21111	4	0.3	97.3
21121	4	0.3	97.6
11123	3	0.2	97.9
12221	3	0.2	98.1
11113	2	0.2	98.3
12121	2	0.2	98.5
21112	2	0.2	98.6
21222	2	0.2	98.8
22333	2	0.2	98.9
11323	1	0.1	99.0
12111	1	0.1	99.1
12123	1	0.1	99.2
12132	1	0.1	99.3
13213	1	0.1	99.3
21122	1	0.1	99.4
21211	1	0.1	99.5
21212	1	0.1	99.6
21223	1	0.1	99.7
22111	1	0.1	99.8
22122	1	0.1	99.8
23223	1	0.1	99.9
Total	1229		100

## Data Availability

The datasets used during the current study are available from the corresponding author on reasonable request.

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
