# Peer review of "Health-Related Quality of Life Norm Data of the Peruvian Adolescents: Results Using the EQ-5D-Y"

_ijerph, 2021, doi:10.3390/ijerph18168735_

Round 1
Reviewer 1 Report
Reviewer report on “Health Related Quality of Life Norm Data of the Peruvian Adolescents: Results Using the EQ-5D-Y” submitted to IJERPH
Overall impression.
The paper under review looks at a potentially important issue. However, the analysis is pretty much undercooked in various aspects. My biggest concern is that this study is descriptive in nature. This kind of information can be found in many statistical yearbooks or government websites. For example, you detected gender and age differences (), so what? Are you (as a researcher) satisfied with knowing there are some gender differences? Wouldn’t you expect these before analyzing your data? Wouldn’t you want to dig a little deeper to see what is causing these differences? Exploring why these differences exist is way more important than merely describing these differences.
Introduction.
One of the contributions of the study is its analysis of Peruvian adolescents. Then, it would be useful to introduce some features about Peru and Peruvian adolescents in the Introduction to better inform the reader. For example, what is unique about Peru and Peruvian adolescents? Do you expect your results to be different than those found in other (LA) countries?
Sampling.
The authors should be more clear about the sampling frame—What’s the underlying population? What’s the sampling design? Are there many refusals? How was participants’ consent provided? Any missing observations? Any replacements of participants? A flowchart showing how the sample was collected would be very helpful for clarifying these.
Analysis.
This study presents only descriptive analyses. More in-depth analyses should be undertaken. For example, you detected gender and age differences (wouldn’t you expect these before analyzing your data), but why do these differences exist? And any measures should be taken to reduce these differences?
Writing.
There are many (relatively minor) language problems. The author should check the manuscript carefully to correct these errors.
Author Response
Comments and Suggestions for Authors
Reviewer report on “Health-Related Quality of Life Norm Data of the Peruvian Adolescents: Results Using the EQ-5D-Y” submitted to IJERPH
Overall impression.
The paper under review looks at a potentially important issue. However, the analysis is pretty much undercooked in various aspects. My biggest concern is that this study is descriptive in nature. This kind of information can be found in many statistical yearbooks or government websites.
Thank you for your comments. We agree to respect your main concern about the descriptive nature of the paper. However, we think that descriptive studies are necessary a very relevant as the first steps of science, at least when the government does not provide that information such as in this case. There is no public information about reference values of the EQ-5D-Y instrument (one of the most relevant for making decisions) of Peruvian adolescents. We have highlighted in the article the importance of this type of study.
For example, you detected gender and age differences (), so what? Are you (as a researcher) satisfied with knowing there are some gender differences? Wouldn’t you expect these before analyzing your data?
To our knowledge, a study headed by Takeru S. has shown population norms with the EQ-5D-Y in Japan. In this study, no differences have been found either on the basis of sex or age. “The EQ-5D-Y values did not differ depending on the age or sex of the children/ adolescents” (1)
This finding is very relevant, since, among other aspects, due to the cultural differences and living conditions of each country, the results (index values) may vary, and it is relevant to be able to study it in each context. In this sense, without having conducted this research in different socio-cultural contexts, it is impossible to know what the results will be.
Wouldn’t you want to dig a little deeper to see what is causing these differences? Exploring why these differences exist is way more important than merely describing these differences.
With respect to the differences between males and females, we had no assumptions about these possible outcomes, our aim was to provide reference values in segmented data to improve its usefulness our aim was to provide reference values in segmented data to improve its usefulness and to understand the reality of the national context. The descriptive nature of the study prevents us from establishing cause-effect relationships and therefore we can’t know what causes these differences or findings. This study is a first step in making it known that differences exist. In order to know the causes, it is necessary to carry out other studies with an adequate design for this purpose. However, it is important to include in the discussion the need to go deeper into the reasons that could explain these differences. We have added a hypothesis in the last part of the discussion section.
Introduction.
One of the contributions of the study is its analysis of Peruvian adolescents. Then, it would be useful to introduce some features about Peru and Peruvian adolescents in the Introduction to better inform the reader. For example, what is unique about Peru and Peruvian adolescents? Do you expect your results to be different than those found in other (LA) countries?
Thank you for your suggestion. We have improved the last part of the introduction to address these questions.
Sampling.
The authors should be clearer about the sampling frame—What’s the underlying population? What’s the sampling design? Are there many refusals? How was participants’ consent provided? Any missing observations? Any replacements of participants? A flowchart showing how the sample was collected would be very helpful for clarifying these.
Thank you for your comments. We have added in the methods section a figure showing the recruitment process.
Fig. 1 Recruitment process.
Analysis.
This study presents only descriptive analyses. More in-depth analyses should be undertaken. For example, you detected gender and age differences (wouldn’t you expect these before analyzing your data), but why do these differences exist? And any measures should be taken to reduce these differences?
This question has been answered in the "overall impression" section of this review.
Writing.
There are many (relatively minor) language problems. The author should check the manuscript carefully to correct these errors.
Done. Thanks.

Reviewer 2 Report
The paper "Health Related Quality of Life Norm Data of the Peruvian Adolescents: Results Using the EQ-5D-Y" is of some interest.
The introduction is unclear, has several repetitions, and should be completely rewritten.
Materials and methods and results are quite clear.
Discussion: some phrases are not in English. The whole chapter needs a substantial revision.
Author Response
The paper "Health-Related Quality of Life Norm Data of the Peruvian Adolescents: Results Using the EQ-5D-Y" is of some interest.
The introduction is unclear, has several repetitions, and should be completely rewritten (
Thank you for your suggestion. The introduction has been revised and improved.
Materials and methods and results are quite clear.
Thank you.
Discussion: some phrases are not in English. The whole chapter needs a substantial revision.
Thank you, we have revised the chapter and changed the non-English parts of the whole document.

Reviewer 3 Report
In this manuscript, Palacios-Cartagena et al. reported the normative values of the EQ-5D-Y questionnaire in Peruvian adolescents. They observed that this self-report instrument was feasible for adolescent students aged 12 to 17 years. Interestingly, they found that male subjects showed better quality of life than females. They also found that as age increases, there seems to be worse HRQOL, this may be to the physical and psychological changes that occur at this stage.
This is a very interesting study with large sample size. The topic is of great importance and emphasizes the concept that health-related quality of life in the adolescent stage is of vital importance. The methods are clearly spelt out and related results are presented effectively.
I have some comments for the authors.
Line 71- Please translate the countries into English
I would recommend adding in the methods section, that EQ-5D-Y has been validated in Spanish
Lines 143.-144 “Being the 13-year-old male age group reporting a mean of 0.931 being the highest in the age group”. Please rephrase this sentence, it is not clear.
The finding that males have a higher quality of life is very meaningful. The authors should discuss some hypotheses on that.
Line 208-210. Please translate the sentence into English
Author Response
Comments and Suggestions for Authors
In this manuscript, Palacios-Cartagena et al. reported the normative values of the EQ-5D-Y questionnaire in Peruvian adolescents. They observed that this self-report instrument was feasible for adolescent students aged 12 to 17 years. Interestingly, they found that male subjects showed better quality of life than females. They also found that as age increases, there seems to be worse HRQOL, this may be to the physical and psychological changes that occur at this stage.
This is a very interesting study with a large sample size. The topic is of great importance and emphasizes the concept that health-related quality of life in the adolescent stage is of vital importance. The methods are clearly spelled out and related results are presented effectively. Thank you for your comments about our work.
I have some comments for the authors.
Line 71- Please translate the countries into English.
Thanks, we have changed the country’s names into English.
I would recommend adding in the methods section, that EQ-5D-Y has been validated in Spanish.
References of the validation of the main instrument and the proxy version have been added to the methods section.
Lines 143.-144 “Being the 13-year-old male age group reporting a mean of 0.931 being the highest in the age group”. Please rephrase this sentence, it is not clear.
Done
The finding that males have a higher quality of life is very meaningful. The authors should discuss some hypotheses on that.
Thanks for your suggestion. We have added a hypothesis in the last part of the discussion section.
Line 208-210. Please translate the sentence into English.
Thank you, we have translated it.

Round 2
Reviewer 1 Report
This paper adds almost NOTHING to the literature. And the authors did very little to improve the paper.
Author Response
This paper adds almost NOTHING to the literature. And the authors did very little to improve the paper.
Thank you very much for reviewing the manuscript and giving us your expert opinion.
We are surprised that in this second review you have marked all the questions in the Review Report Form "should be improved" when in the previous review you only marked 2 of the 5 questions "should be improved", could you tell us why this is so?
Manuscripts with normative data are quite common in the scientific literature in prestigious journals with high impact in the JCR for instance some of the latest manuscripts are:
- Marten, O.; Greiner, W. EQ-5D-5L reference values for the German general elderly population. Health and Quality of Life Outcomes 2021, 19, 1-11.
- Gutierrez-Delgado, C.; Galindo-Suárez, R.-M.; Cruz-Santiago, C.; Shah, K.; Papadimitropoulos, M.; Feng, Y.; Zamora, B.; Devlin, N. EQ-5D-5L Health-State Values for the Mexican Population. Applied Health Economics and Health Policy 2021, 1-10.
- Yao, Q.; Liu, C.; Zhang, Y.; Xu, L. Population norms for the EQ-5D-3L in China derived from the 2013 National Health Services Survey. Journal of global health 2021, 11.
- Encheva, M.; Djambazov, S.; Vekov, T.; Golicki, D. EQ-5D-5L Bulgarian population norms. Eur J Health Econ 2020, 21, 1169-1178, doi:10.1007/s10198-020-01225-5.
According to the scientific literature, although these studies are descriptive, they have several important utilities:
Data obtained from this questionnaire in a general population (normative values) allows comparisons of HRQoL with other pathological populations, providing information on specific domains that differ between subjects with or without symptoms to address during treatment or compare patient profiles with particular conditions, similar age group or gender, aiding health policy development and planning [1]
We have answered in detail everything you asked us in your first review. In this second review you do not indicate anything specific that needs to be improved, so we do not know what you are referring to. Here are the answers to all the comments you made in your first review. If you could specify what we need to improve, we would appreciate it.
Reviewer report on “Health-Related Quality of Life Norm Data of the Peruvian Adolescents: Results Using the EQ-5D-Y” submitted to IJERPH
Overall impression.
The paper under review looks at a potentially important issue. However, the analysis is pretty much undercooked in various aspects. My biggest concern is that this study is descriptive in nature. This kind of information can be found in many statistical yearbooks or government websites.
Thank you for your comments. We agree to respect your main concern about the descriptive nature of the paper. However, we think that descriptive studies are necessary a very relevant as the first steps of science, at least when the government does not provide that information such as in this case. There is no public information about reference values of the EQ-5D-Y instrument (one of the most relevant for making decisions) of Peruvian adolescents. We have highlighted in the article the importance of this type of study.
For example, you detected gender and age differences (), so what? Are you (as a researcher) satisfied with knowing there are some gender differences? Wouldn’t you expect these before analyzing your data?
To our knowledge, a study headed by Takeru S. has shown population norms with the EQ-5D-Y in Japan. In this study, no differences have been found either on the basis of sex or age. “The EQ-5D-Y values did not differ depending on the age or sex of the children/ adolescents” (1)
This finding is very relevant, since, among other aspects, due to the cultural differences and living conditions of each country, the results (index values) may vary, and it is relevant to be able to study it in each context. In this sense, without having conducted this research in different socio-cultural contexts, it is impossible to know what the results will be.
Wouldn’t you want to dig a little deeper to see what is causing these differences? Exploring why these differences exist is way more important than merely describing these differences.
With respect to the differences between males and females, we had no assumptions about these possible outcomes, our aim was to provide reference values in segmented data to improve its usefulness our aim was to provide reference values in segmented data to improve its usefulness and to understand the reality of the national context. The descriptive nature of the study prevents us from establishing cause-effect relationships and therefore we can’t know what causes these differences or findings. This study is a first step in making it known that differences exist. In order to know the causes, it is necessary to carry out other studies with an adequate design for this purpose. However, it is important to include in the discussion the need to go deeper into the reasons that could explain these differences. We have added a hypothesis in the last part of the discussion section.
Introduction.
One of the contributions of the study is its analysis of Peruvian adolescents. Then, it would be useful to introduce some features about Peru and Peruvian adolescents in the Introduction to better inform the reader. For example, what is unique about Peru and Peruvian adolescents? Do you expect your results to be different than those found in other (LA) countries?
Thank you for your suggestion. We have improved the last part of the introduction to address these questions.
Sampling.
The authors should be clearer about the sampling frame—What’s the underlying population? What’s the sampling design? Are there many refusals? How was participants’ consent provided? Any missing observations? Any replacements of participants? A flowchart showing how the sample was collected would be very helpful for clarifying these.
Thank you for your comments. We have added in the methods section a figure showing the recruitment process.
Fig. 1 Recruitment process.
Analysis.
This study presents only descriptive analyses. More in-depth analyses should be undertaken. For example, you detected gender and age differences (wouldn’t you expect these before analyzing your data), but why do these differences exist? And any measures should be taken to reduce these differences?
This question has been answered in the "overall impression" section of this review.
Writing.
There are many (relatively minor) language problems. The author should check the manuscript carefully to correct these errors.
Done. Thanks.
- Kreimeier, S.; Greiner, W. EQ-5D-Y as a health-related quality of life instrument for children and adolescents: the instrument's characteristics, development, current use, and challenges of developing its value set. Value in Health 2019, 22, 31-37.

Reviewer 2 Report
the paper is acceptable for publication in its present form
Author Response
Thank you very much for your time and suggestions for improving the manuscript.